# Prognostic value of 11C-methionine volume-based PET parameters in IDH wild type glioblastoma

**Bart R. J. van Dijken**[1]*, **Alfred O. Ankrah**[2], **Gilles N. Stormezand**[2], **Rudi A. J. O. Dierckx**[1,2], **Peter Jan van Laar**[1,3], **Anouk van der Hoorn**[1]

**1** Department of Radiology, University Medical Center Groningen, University of Groningen, Groningen, the Netherlands, **2** Department of Nuclear Medicine and Molecular Imaging, University Medical Center Groningen, University of Groningen, Groningen, the Netherlands, **3** Department of Radiology, Zorggroep Twente, Almelo and Hengelo, the Netherlands

* b.r.j.van.dijken@umcg.nl

**Data Availability Statement:** We have uploaded the minimal (anonymized) data set as supporting information.

## Abstract

### Purpose

$^{11}$C-Methionine ($^{11}$C-MET) PET prognostication of isocitrate dehydrogenase (IDH) wild type glioblastomas is inadequate as conventional parameters such as standardized uptake value (SUV) do not adequately reflect tumor heterogeneity. We retrospectively evaluated whether volume-based parameters such as metabolic tumor volume (MTV) and total lesion methionine metabolism (TLMM) outperformed SUV for survival correlation in patients with IDH wild type glioblastomas.

### Methods

Thirteen IDH wild type glioblastoma patients underwent preoperative $^{11}$C-MET PET. Both SUV-based parameters and volume-based parameters were calculated for each lesion. Kaplan-Meier curves with log-rank testing and Cox regression analysis were used for correlation between PET parameters and overall survival.

### Results

Median overall survival for the entire cohort was 393 days. MTV (HR 1.136, p = 0.007) and TLMM (HR 1.022, p = 0.030) were inversely correlated with overall survival. SUV-based $^{11}$C-MET PET parameters did not show a correlation with survival. In a paired analysis with other clinical parameters including age and radiotherapy dose, MTV and TLMM were found to be independent factors.

### Conclusions

MTV and TLMM, and not SUV, significantly correlate with overall survival in patients with IDH wild type glioblastomas. The incorporation of volume-based $^{11}$C-MET PET parameters may lead to a better outcome prediction for this heterogeneous patient population.

**Funding:** The author(s) received no specific funding for this work.

**Competing interests:** The authors have declared that no competing interests exist.

## Introduction

The use of amino acid PET such as L-methyl-[11]C-methionine ([11]C-MET) for prognostication of gliomas has recently been recommended [1]. [11]C-MET PET has an excellent tumor-to-background discrimination and provides supplementary metabolic information in addition to MRI [2].

Survival and treatment strategies differ significantly among glioma patients, necessitating reliable preoperative prognostication [3]. Gliomas without isocitrate dehydrogenase (IDH) mutation, including the WHO grade 4 glioblastoma, are now recognized as the most malignant subtype [4]. These IDH wild type gliomas are heterogeneous tumors characterized by a more aggressive and infiltrative nature than IDH mutant gliomas [5]. Glioblastomas IDH wild type WHO grade 4 are defined due to either aggressive histological features such as microvascular proliferation and necrosis, or by specific molecular markers such as TERT promoter mutation, EGFR amplification, and chromosome 7 gain/chromosome 10 loss [4]. Outcome in glioblastomas is generally poor.

The prognostic use of [11]C-MET PET in gliomas has previously been explored but showed inconclusive results [2]. Studies were usually limited to conventional parameters such as standardized uptake value (SUV), which do not reliably detect the heterogeneity of IDH wild type glioblastomas [6]. Recently, volume-based parameters have been introduced, possibly better reflecting the heterogeneity of IDH wild type glioblastomas [7, 8]. Volume-based parameters such as metabolic tumor volume (MTV) and total lesion methionine metabolism (TLMM), a combination of MTV and SUVmean, provide additional information about the tumor extent. The aim of this study was to investigate the value of conventional and volume-based [11]C-MET PET parameters for prognostication in IDH wild type glioblastomas.

## Methods

We performed a retrospective search among all [11]C-MET PET examinations in our tertiary university hospital between 2011–2018. Inclusion criteria were glioblastoma, IDH wild type, WHO grade 4 according to the 2021 WHO guidelines, confirmed by biopsy or surgical resection ≤3 months after PET examination [4]. Exclusion criteria were previous cranial surgery or cerebral irradiation, known other primary tumor, and pediatric patients (<18 years). The study was approved by the institutional review board and the need for written informed consent was waived.

The tracer was prepared as previously published [9] and imaging was performed in accordance with the 2006 procedure guidelines of the European Association of Nuclear Medicine for brain tumor imaging using labelled amino acid analogues [10]. Patients underwent static PET imaging 20 minutes after intravenous injection of 200 MBq (range: 180–220 MBq) [11]C-MET in one bed position of five minutes. Images were acquired on a Biograph mCT PET/CT system (Siemens/CTI, Knoxville, TN, USA) with 2 mm spatial resolution. The images were reconstructed using Truex + TOF with three iterations and 21 subsets in a 400 x 400 matrix size (zoom 1.0) and a 2 mm Gaussian filter.

Volumes of interest (VOI) were defined using Syngo.via (Siemens Medical Solutions Inc., Knoxville, TN, USA). A circle large enough to include all visual uptake was manually drawn by one researcher (BD) and checked by a nuclear physician with 3 years of experience (GS). The VOI were semi-automatically defined using a 40% threshold of SUVmax [6], which is standard value for delineation used in our research group and leads to good tumor definition when compared to visual inspection. Syngo.via allowed for extraction of the following PET parameters within the VOI: SUVmax, SUVpeak, SUVmean, MTV and TLMM (Fig 1). TLMM was

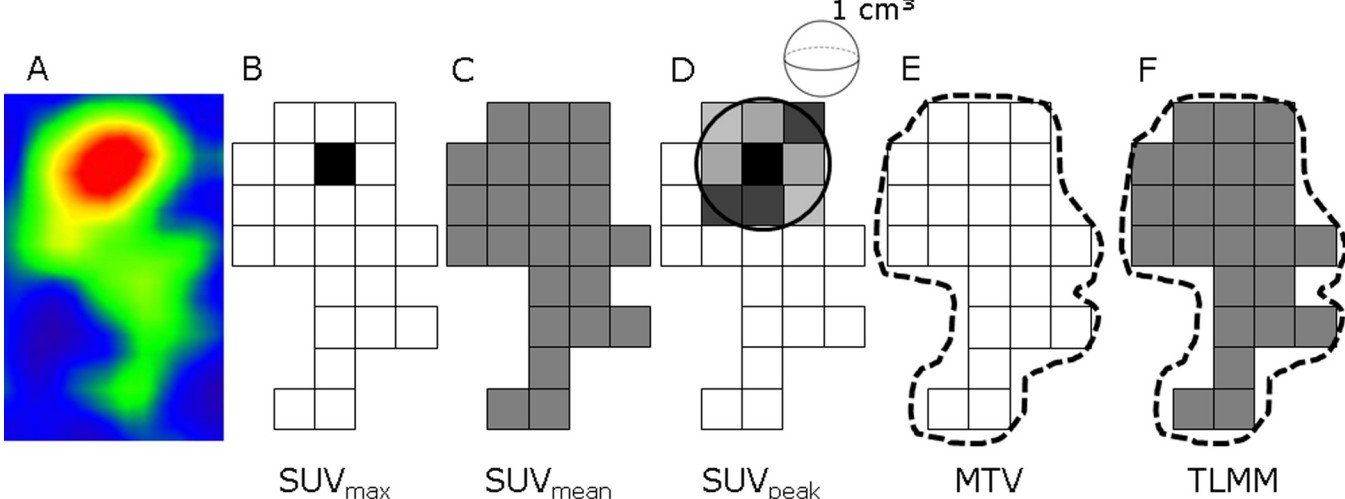

**Fig 1. Schematic overview of metabolic and volume-based $^{11}$C-MET PET parameters.** (A) Snapshot of lesion with positive uptake of 11C-MET. (B) SUVmax is defined as voxel with highest uptake within the lesion. (C) SUVmean is an average of all SUV values from included voxels within the lesion. (D) SUVpeak is defined as the average value within a 1cm$^3$ sphere surrounding the SUVmax. Volume-based parameters are shown in E-F. (E) MTV is the volume of increased uptake within the lesion borders. (F) For TLMM the SUVmean and MTV within lesion borders are combined and multiplied to provide both metabolic and volume-based information. TLMM is automatically calculated and extracted within the Syngo.via software. Abbreviations: MTV = metabolic tumor volume, SUV = standardized uptake value, TLMM = total lesion methionine metabolism.

automatically calculated in Syngo.via by multiplying SUVmean and MTV within the lesion border, in an equal manner as the Total Lesion Glycolysis (TLG) in FDG imaging.

Statistics were done in SPSS (IBM corp, version 25.0, Armonk, NY). Log-rank testing after dichotomization based on median values and Cox regression analysis were used for univariate survival analysis. The influence of various clinical variables (age, tumor grade, extent of resection and radiation dose) was also determined since the sample size was insufficient for multivariate analysis.

## Results

A total of 13 patients with a median age of 57 years (range 28–70) and of whom 62% were male were included in the study. Adjuvant treatment consisted of radiotherapy plus or minus chemotherapy. One patient refused to undergo additional treatment after biopsy and one patient had to discontinue radiotherapy due to adverse events after a dose of 32.4 Gy. General characteristics are shown in Table 1.

Median survival was 393 days. The results of univariate analysis for the different PET parameters are shown in Table 2. MTV and TLMM were the only parameters that significantly correlated with overall survival (hazard ratio [HR] 1.136 (95%CI 1.035–1.246), $p$ = 0.007 and HR 1.022 (95%CI 1.002–1.043), $p$ = 0.030, respectively). Kaplan-Meier curves for MTV and TLMM are shown in Fig 2. Paired analysis with other clinical parameters demonstrated that MTV and TLMM were correlated with survival, independent of age, extent of resection or radiation dose. SUV based parameters were not correlated with survival (Fig 3).

## Discussion

Volume-based $^{11}$C-MET PET parameters were significantly correlated with survival in this small retrospective study among IDH wild type glioblastoma patients, while SUV based

**Table 1. General characteristics of included patients.**

| Patient | Gender | Age | WHO Grade | EOR | Treatment | RT dose | Survival |
|---|---|---|---|---|---|---|---|
| 1 | F | 70 | GBM, gr 4* | GTR | RT | 57.6 | 1325 |
| 2 | F | 63 | GBM, gr 4 | STR | C+RT | 60 | 195‡ |
| 3 | M | 28 | GBM, gr 4* | B | C+RT | 60 | 32‡ |
| 4 | F | 64 | GBM, gr 4* | B | C+RT | 32.4† | 221 |
| 5 | F | 57 | GBM, gr 4 | B | C+RT | 60 | 418 |
| 6 | M | 65 | GBM, gr 4 | B | None | - | 266 |
| 7 | M | 59 | GBM, gr 4 | B | C+RT | 60 | 200 |
| 8 | M | 44 | GBM, gr 4 | GTR | C+RT | 60 | 277 |
| 10 | M | 47 | GBM, gr 4 | B | C+RT | 60 | 171 |
| 11 | M | 61 | GBM, gr 4 | B | C+RT | 60 | 618 |
| 12 | M | 52 | GBM, gr 4 | B | C+RT | 60 | 469 |
| 13 | M | 67 | GBM, gr 4* | B | RT | 59.4 | 688 |
| 15 | F | 60 | GBM, gr 4* | B | C+RT | 59.4 | 237 |

* defined by EGFR amplification

†discontinued radiotherapy due to adverse events.

parameters were not. TLMM, combining both volume-based and metabolic information, can thus be potentially employed for prognostication in IDH wild type glioblastomas.

Conventional SUV measures are the most commonly used quantitative PET parameters. Several studies have investigated the use of [11]C-MET SUV measures for glioma prognostication, with inconclusive results [2, 11–14]. SUV measures, however, do not provide information about tumor heterogeneity, nor do they visualize the extent of the lesion. In the current study we did not find a correlation between SUV measures and survival in IDH wild type glioblastomas, while volume-based parameters did demonstrate this correlation.

Our findings are in accordance with another [11]C-MET PET study which also demonstrated that a larger MTV and TLMM were correlated with lower survival [7]. However, important limitations of that study were the inclusion of oligodendrogliomas and lack of correcting for IDH mutation status, whilst it is known that both survival and [11]C-MET uptake values significantly differ between IDH wild type and IDH-1 gliomas [15, 16]. Furthermore, IDH wild type glioblastomas, are currently seen as separate entity and have a poorer prognosis than IDH-1 gliomas [4]. The prognostic role of volume-based parameters in gliomas have also been demonstrated for other amino acid PET tracers, such as O-(2-[18F]fluoroethyl)-l-tyrosine (18F-FET) [17–19]. for oncology prognostication has also been demonstrated in other populations [6, 20–23].

**Table 2. Univariate analysis of [11]C-MET PET parameters and survival.**

| Parameter | HR (95%CI) | p value |
|---|---|---|
| SUVmax | 0.818 (0.583–1.148) | 0.246 |
| SUVpeak | 0.833 (0.549–1.263) | 0.389 |
| SUVmean | 0.669 (0.366–1.221) | 0.190 |
| MTV | 1.136 (1.035–1.246) | 0.007 |
| TLMM | 1.022 (1.002–1.043) | 0.030 |

Abbreviations: HR = hazard ratio, MTV = metabolic tumor volume, SUV = standard uptake value, TLMM = total lesion methionine metabolism.

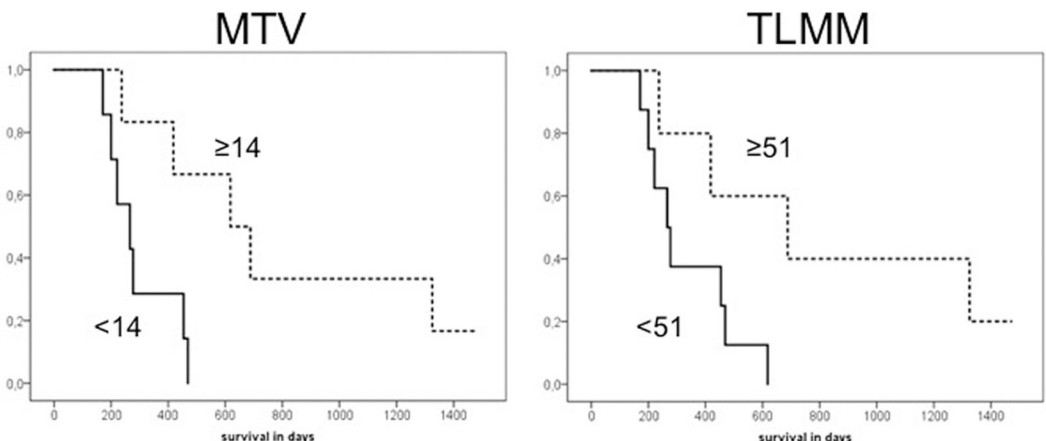

**Fig 2. Kaplan-Meier curves for MTV and TLMM.** Survival curves for MTV and TLMM. Dichotomisation of the data based on the median ($\geq$14 for MTV and $\geq$51 for TLMM). The group with higher values is displayed with the dotted line while the group with lower values is displayed with the bold line. The survival difference between patients with higher and lower values was statistically significant for both MTV (p = 0.014) and TLMM (p = 0.041). Abbreviations: MTV = metabolic tumor volume, TLMM = total lesion methionine metabolism.

Volume-based PET parameters are strongly dependent on the used delineation method [6]. This was recently demonstrated for $^{18}$F-FDG PET in non-small cell lung cancer patients, where MTV significantly differed with chosen delineation method [6]. According to the authors of that study, the approach that reached best agreement was 40% of SUVmax, corresponding to the method used in the present study [6]. However, these results have not yet been validated for $^{11}$C-MET PET. Moreover, it is not known if these results can be translated to gliomas. More research should therefore be aimed at establishing a robust delineation method for $^{11}$C-MET PET in gliomas.

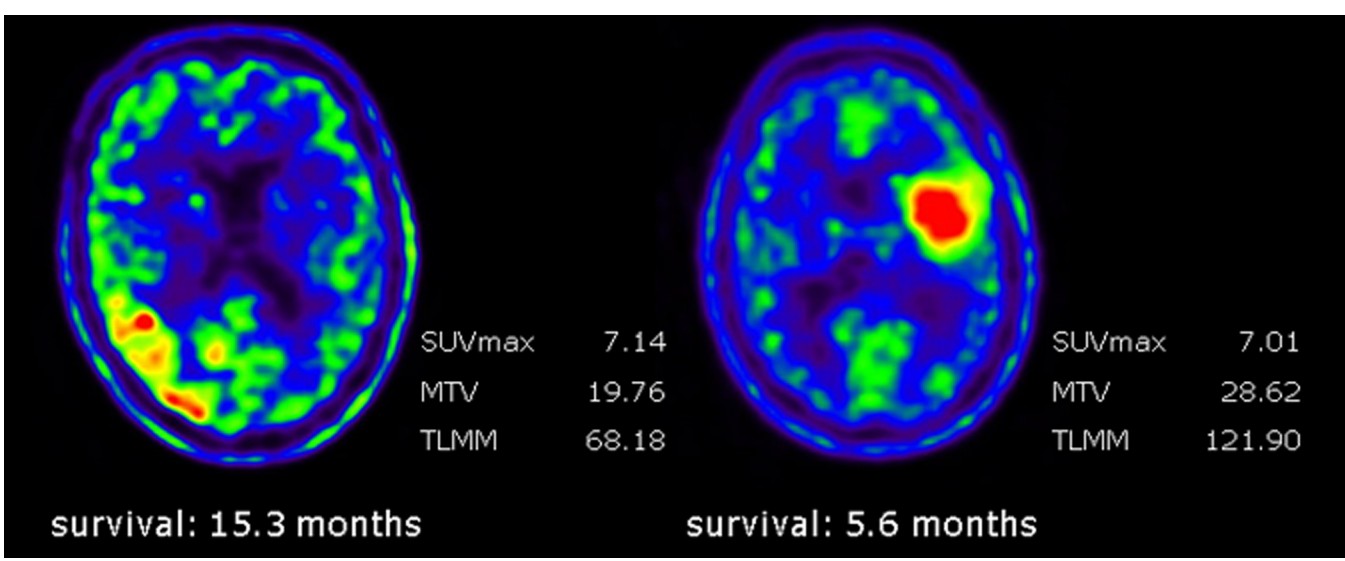

**Fig 3. Differences in metabolic and volumetric parameters in two glioblastoma patients.** Left image: 52-year-old male with right temporoparietal glioblastoma. Right image: 47-year-old male with left frontal glioblastoma. Despite relatively similar SUVmax values, there was a significant difference in survival between the two patients. The tumor of the patient on the right side demonstrated higher MTV and TLMM values than the tumor on the left side, corresponding to the lower survival of this patient.

The most important limitation of this study is the small sample size. Due to this, a comprehensive multivariate analysis was not possible. Nevertheless, in paired analysis with other clinical parameters TLMM demonstrated to be independently correlated to survival. Another limitation is the retrospective nature of this study. Due to this, molecular markers, such as TERT promoter mutation, EGFR amplification, and gain of chromosome 7 with loss of chromosome 10 status was not always known, contributing to the smaller sample size. Our results should thus be seen as an encouragement for larger prospective studies focusing on the prognostic role of volume-based [11]C-MET PET parameters. At present, a prospective study on 11C-MET PET for treatment evaluation in glioma patients is ongoing at out tertiary center (Netherlands Trial Register number NL6536) which will also address the utility of volume-based parameters for prognostication.

## Conclusion

Volume-based [11]C-MET PET parameters MTV and TLMM were significantly correlated with overall survival in patients with IDH wild type glioblastomas. Volume-based parameters could play a future role in tailoring treatment decisions in patients with IDH wild type glioblastoma.

## Supporting information

**S1 File.**
(PDF)

**S2 File.**
(PDF)

**S1 Data.**
(XLSX)

## Author Contributions

**Conceptualization:** Bart R. J. van Dijken, Alfred O. Ankrah, Rudi A. J. O. Dierckx, Peter Jan van Laar, Anouk van der Hoorn.

**Data curation:** Bart R. J. van Dijken.

**Formal analysis:** Bart R. J. van Dijken, Alfred O. Ankrah, Gilles N. Stormezand, Anouk van der Hoorn.

**Investigation:** Bart R. J. van Dijken, Alfred O. Ankrah, Anouk van der Hoorn.

**Methodology:** Bart R. J. van Dijken, Alfred O. Ankrah, Gilles N. Stormezand, Rudi A. J. O. Dierckx, Peter Jan van Laar, Anouk van der Hoorn.

**Project administration:** Bart R. J. van Dijken, Anouk van der Hoorn.

**Supervision:** Gilles N. Stormezand, Rudi A. J. O. Dierckx, Peter Jan van Laar, Anouk van der Hoorn.

**Writing – original draft:** Bart R. J. van Dijken, Anouk van der Hoorn.

**Writing – review & editing:** Bart R. J. van Dijken, Alfred O. Ankrah, Gilles N. Stormezand, Rudi A. J. O. Dierckx, Peter Jan van Laar, Anouk van der Hoorn.

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
