## [Decision Letter · Decision Letter 0]

1 Dec 2021

PONE-D-21-28633Prognostic value of 11C-methionine volume-based PET parameters in IDH wild type gliomasPLOS ONE

Dear Dr. Van Dijken,

Thank you for submitting your manuscript to PLOS ONE. After careful consideration, we feel that it has merit but does not fully meet PLOS ONE’s publication criteria as it currently stands. Therefore, we invite you to submit a revised version of the manuscript that addresses the points raised during the review process. Please submit your revised manuscript by Jan 15 2022 11:59PM. If you will need more time than this to complete your revisions, please reply to this message or contact the journal office at plosone@plos.org. Please include the following items when submitting your revised manuscript:A rebuttal letter that responds to each point raised by the academic editor and reviewer(s). You should upload this letter as a separate file labeled 'Response to Reviewers'.A marked-up copy of your manuscript that highlights changes made to the original version. You should upload this as a separate file labeled 'Revised Manuscript with Track Changes'.An unmarked version of your revised paper without tracked changes. You should upload this as a separate file labeled 'Manuscript'.

We look forward to receiving your revised manuscript.

Kind regards,

Kevin Camphausen

Academic Editor

PLOS ONE

Journal Requirements:

Reviewers' comments:

Reviewer's Responses to Questions

**Comments to the Author**

1. Is the manuscript technically sound, and do the data support the conclusions?

Reviewer #1: Yes

Reviewer #2: No

2. Has the statistical analysis been performed appropriately and rigorously? 

Reviewer #1: Yes

Reviewer #2: N/A

3. Have the authors made all data underlying the findings in their manuscript fully available?

Reviewer #1: Yes

Reviewer #2: No

4. Is the manuscript presented in an intelligible fashion and written in standard English?

Reviewer #1: Yes

Reviewer #2: Yes

5. Review Comments to the Author

Reviewer #1: This paper on the prognostic value of 11C-methionine volume-based PET parameters in IDH wild type

gliomas is timely and deals with an important imaging niche (11C-methionine PET) and specifically the need for acquired parameters to become increasingly clinically meaningful (SUV vs. volume based ). It is to be commended on not simply going after the SUV and in its emphasis on focusing on wild type IDH mutant gliomas. However it deals with only 15 patients (which the authors acknowledge as a limitation) and a larger cohort as well as validation with larger scale data perhaps from another institution and/or in conjunction with large scale data bases (TCGA/BRATS) would add weight to the findings and advance the field.

Reviewer #2: Prognostic value of 11C-methionine volume-based PET parameters in IDH wild type gliomas.

This article aims to demonstrate the prognostic value of 11C-methionine (11C-met) PET in the survival of IDH wild type glioma. While previous trials of 11C-met trials have proven inconclusive, the authors of this paper employ volume-based parameters such as metabolic tumor volume (MTV) and total lesion methionine metabolism (TLMM) in an attempt to improve prognostic accuracy. This is an important question, as better non-invasive methods of diagnosing glioma are desperately needed. Unfortunately, this paper has several structural issues that cannot be fixed and prevent its publication:

1) The sample size of this study is too small and heterogenous to make any conclusions. While it is true that IDH wild type gliomas do generally carry worse prognosis that IDH mutants, grade IV, grade III and grade II gliomas represent different pathologies with different behaviors and responses to treatment.

2) The definition of MTV and TLMM are unclear. In the legend of Figure 1, you state that SUVmean and MTV are combined and multiplied to obtain TLMM however the diagram accompanied does not clearly demonstrate how this is applied in practice.

3) The Kaplan Meier curves from figure 2 is unclear. Are the values based on volume or metabolic activity?

4) While table 1 provides individual data on extent of resection, age and radiation dose, graphical data may better delineate how these factors impact survival

6. PLOS authors have the option to publish the peer review history of their article (what does this mean?). If published, this will include your full peer review and any attached files.

Reviewer #1: No

Reviewer #2: No

---

## [Author Response · Author response to Decision Letter 0]

26 Jan 2022

Response to the reviewers

Reviewer #1: 

Comment: 

This paper on the prognostic value of 11C-methionine volume-based PET parameters in IDH wild type gliomas is timely and deals with an important imaging niche (11C-methionine PET) and specifically the need for acquired parameters to become increasingly clinically meaningful (SUV vs. volume based ). It is to be commended on not simply going after the SUV and in its emphasis on focusing on wild type IDH mutant gliomas. However, it deals with only 15 patients (which the authors acknowledge as a limitation) and a larger cohort as well as validation with larger scale data perhaps from another institution and/or in conjunction with large scale data bases (TCGA/BRATS) would add weight to the findings and advance the field.

Answer: 

We would like to thank the reviewer for critically reading our manuscript and commending us on the scope of this study. The small sample size indeed is one of the most important limitations of our study, which we also acknowledge in the discussion section. A larger study cohort would definitely strengthen the study. However, due to the retrospective nature of our study and the study niche (11C-methionine PET in IDH wild type gliomas) this was not possible. We thank the reviewer for the suggestion of using large scale databases to increase the number of subjects. Unfortunately, the TCGA and BRATS databases do not include preoperative 11C-methionine PET data. Therefore, producing a larger cohort for our study at this stage was not deemed feasible. 

Despite small sample size, the found correlation of the volume-based parameters MTV and TLMM with survival was statistically significant and TLMM was even suggested to be an independent factor in paired analysis with other clinical parameters. We feel that our results should thus be seen as an encouragement for larger prospective studies focusing on the prognostic role of volume-based 11C-MET PET parameters to confirm our results. At a present, a prospective study on 11C-MET PET treatment evaluation in glioma patients is undertaken at our tertiary hospital (Netherlands Trial Register number NL6536; https://www.trialregister.nl/trial/6536), which includes preoperative imaging as well. The value of volume-based parameters for prognostication (and treatment response assessment) will also be a scope of this study. A sentence to highlight this was added to the final paragraph of the discussion. 

Reviewer #2: 

Comment:

Prognostic value of 11C-methionine volume-based PET parameters in IDH wild type gliomas. This article aims to demonstrate the prognostic value of 11C-methionine (11C-met) PET in the survival of IDH wild type glioma. While previous trials of 11C-met trials have proven inconclusive, the authors of this paper employ volume-based parameters such as metabolic tumor volume (MTV) and total lesion methionine metabolism (TLMM) in an attempt to improve prognostic accuracy. This is an important question, as better non-invasive methods of diagnosing glioma are desperately needed. Unfortunately, this paper has several structural issues that cannot be fixed and prevent its publication.

Answer:

We thank the reviewer for the thorough assessment of our manuscript. We agree with the reviewer that establishing more accurate non-invasive methods for the diagnosis and prognostication of IDH wild type gliomas is crucial. This study investigated the value of volume-based 11C-MET PET parameters in addition to SUV, as it is known that SUV does not reliably detect the heterogeneity of IDH wild type gliomas.

Comment:

The sample size of this study is too small and heterogenous to make any conclusions. While it is true that IDH wild type gliomas do generally carry worse prognosis that IDH mutants, grade IV, grade III and grade II gliomas represent different pathologies with different behaviors and responses to treatment.

Answer:

We understand the concern of the reviewer about the small sample size of this study. We also acknowledge this limitation in the last paragraph of the discussion section. Nevertheless, our results were statistically significant and showed that volume-based parameters were correlated with overall survival while conventional parameters were not. Volume-based parameters could play a future role in tailoring treatment decisions in patients with IDH wild type glioblastomas, but we agree that future studies are necessary to confirm our results. We also refer to our answer to reviewer #1. 

Historically, the pathological grade was one of the most important predictors of survival in glioma patients. However, with recent guideline updates, molecular markers now play a prominent role. Due to the poor prognosis, IDH wild type tumors are seen as a different entity than IDH mutated gliomas. In addition, certain molecular markers (TERT promoter mutation, EGFR gene amplification, +7/-10 chromosome changes) now define glioblastoma IDH wild type WHO grade 4, even if histologically the tumor is of lower grade. We agree with the reviewer that the heterogeneity was a point of concern. The reviewer’s comment prompted us to critically re-review our dataset according to the most recent guidelines (2021 WHO Classification of Tumors of the Central Nervous System, Louis et al, Neuro-Oncology, 2021). EGFR gene amplification was seen in a subset of patients in absence of glioblastoma grade 4 defining features such as microvascular proliferation or necrosis. We updated the grades of these patients in table 1. Two patients did not demonstrate any glioblastoma-defining molecular markers, and were therefore deleted from the analysis. In agreement with the newest guidelines, we have also changed the terminology throughout the manuscript to IDH wild type glioblastoma, WHO grade 4. 

Comment:

The definition of MTV and TLMM are unclear. In the legend of Figure 1, you state that SUVmean and MTV are combined and multiplied to obtain TLMM however the diagram accompanied does not clearly demonstrate how this is applied in practice.

Answer:

We agree that the legend of figure 1 was somewhat brief. TLMM is the product of SUVmean and MTV within the lesion border. This parameter can be of course be calculated by hand, but most software allow for automatic calculation and extraction of this TLMM; equal to Total Lesion Glycolysis (TLG) in FDG PET imaging. To clarify the definitions of MTV and TLMM to the reader, we have made the following changes to the text and the legend of figure 1:

Method section, final sentence of paragraph 3: TLMM was automatically calculated in Syngo.via by multiplying SUVmean and MTV within the lesion border, in an equal manner as the Total Lesion Glycolysis (TLG) in FDG imaging.

Legend of figure 1: (…) (E) MTV is the volume of increased uptake within the lesion borders. (F) For TLMM the SUVmean and MTV within lesion borders are combined and multiplied to provide both metabolic and volume-based information. TLMM is automatically calculated and extracted within the Syngo.via software.

Comment:

The Kaplan Meier curves from figure 2 is unclear. Are the values based on volume or metabolic activity? 

Answer:

We agree with the reviewer that the Kaplan Meier curves from figure 2 are unclear. Therefore, to increase clarity, we included the thresholds (median values) of both MTV (�14) and TLMM (�51) for which the data were dichotomized. The group with higher values is displayed with the dotted line while the group with lower values is displayed with the bold line as can be read in the legend. The values for MTV are based on volume and TLMM is a dimensionless parameter which combines the volumetric and metabolic information and is thus not expressed in a particular unit. 

Comment:

4) While table 1 provides individual data on extent of resection, age and radiation dose, graphical data may better delineate how these factors impact survival.

Answer:

In accordance with the reviewer’s suggestion we have included the paired analyses between MTV/TLMM and clinical parameters, as well as graphical data (Kaplan Meier curves) for the impact of clinical parameters on survival in supplementary data 1 and 2, respectively.

---

## [Editor Report · Decision Letter 1]

10 Feb 2022

Prognostic value of 11C-methionine volume-based PET parameters in IDH wild type glioblastoma

PONE-D-21-28633R1

Dear Dr. Van Dijken,

We’re pleased to inform you that your manuscript has been judged scientifically suitable for publication and will be formally accepted for publication once it meets all outstanding technical requirements.

Kind regards,

Kevin Camphausen

Academic Editor

PLOS ONE
---

## [Editor Report · Acceptance letter]

14 Feb 2022

PONE-D-21-28633R1 

Prognostic value of 11C-methionine volume-based PET parameters in IDH wild type glioblastoma 

Dear Dr. Van Dijken:

I'm pleased to inform you that your manuscript has been deemed suitable for publication in PLOS ONE. Congratulations! Your manuscript is now with our production department. 

Kind regards, 

on behalf of

Dr. Kevin Camphausen 

Academic Editor

PLOS ONE